# Gender-based market constraints to informal fish retailing: Evidence from analysis of variance and linear regression

Seamus Murphy[1]☯*, Diksha Arora[2]☯, Froukje Kruijssen[3]☯, Cynthia McDougall[4]☯, Paula Kantor[4]☯

1 School of Natural Sciences, University of Bangor, Bangor, Wales, 2 Department of Economics, Colorado State University, Fort Collins, Colorado, United States of America, 3 Department of Sustainable Economic Development & Gender, KIT Royal Tropical Institute, Amsterdam, the Netherlands, 4 Global Gender Research Cluster, WorldFish, Penang, Malaysia

☯ These authors contributed equally to this work.
* seamusrobertmurphy@gmail.com

**Data Availability Statement:** All relevant data are within the manuscript and its Supporting Information files.

## Abstract

Over the last decade, Egypt's aquaculture sector has expanded rapidly, which has contributed substantially to per capita fish supply, and the growth of domestic fish markets and employment across the aquaculture value chain. Despite the growing importance of aquaculture sector in Egyptian labour force, only a few studies have explored the livelihoods of Egypt's women and men fish retailers. Even fewer studies have examined gender-based market constraints experienced by these informal fish retailers. This study uses sex-disaggregated data collected in 2013 in three governorates of Lower Egypt to examine the economic and social constraints to scale of enterprises between women (n = 162) and men informal fish retailers (n = 183). Specifically, we employ linear regression method to determine the correlates of enterprise performance. We found that both women and men retailers in the informal fish market earn low profits and face livelihood insecurities. However, women's enterprise performance is significantly lower than that of men even after controlling for individual socio-economic and retailing characteristics. Specifically, the burden of unpaid household work and lack of support therein impedes women's ability to generate higher revenues. These findings strengthen the argument for investing in understanding how gender norms and attitudes affect livelihood options and outcomes. This leads to recommendations on gender-responsive interventions that engage with both men and women and enhance the bargaining power and collective voice of fish retailers.

## Introduction

Aquaculture has become the world's fastest growing food sector in the world [1,2]. In Egypt, fish farming has expanded rapidly, which has contributed to domestic fish supply more than doubling over the last ten years [3]. This has led to increasing employment in retail markets [4–7]. According to a recent value chain study by Nasr-Allah et al. [8], aquaculture supply

**Funding:** This work was undertaken as part of the CGIAR Research Program on Fish Agri-Food Systems (FISH) led by WorldFish. The program is supported by contributors to the CGIAR Trust Fund. Additional funding support for this work was provided by the Swiss Agency for Development and Cooperation (https://www.eda.admin.ch/sdc). The funder provided support in the form of salaries for authors SM, CM, FK, but did not have any additional role in the study design, data collection and analysis, decision to publish, or preparation of the manuscript. The specific roles of these authors are articulated in the 'author contributions' section. The funders had no role in the study design, data collection and analysis, decision to publish, or preparation of the manuscript.

**Competing interests:** The authors declare that they have no known competing financial interests or personal relationships that could have appeared to influence the work reported in this paper. The work was commissioned by the Swiss Agency for Development Cooperation and the WorldFish Centre. We declare that this does not alter our adherence to PLOS ONE policies on sharing data and materials.

markets sustain 19 full-time employment opportunities (FTE) for every 100 tonnes of fish produced, while highest rates of employment are generated in retail sectors (9.7 FTE/100t). It is among retail markets where substantial numbers of women traders operate as informal and unlicensed retailers [9].

However, research indicates that the benefits from fish trade tend to be unequally distributed between women and men retailers, particularly in terms of sales volumes and gross profits [10–12]. In Egypt, studies suggest that this is related to challenges in accessing credit and supply markets [13], and market services [7]. Among informal fish markets, the insecurity of marketplace tenure and adverse working conditions such as absence of running water, electricity and limited cold chain facilities add pressure on retailing operations. Such challenges risk the quality and value of fish sold throughout the day, which is reflected in price decline between morning and evening sales [13]. Therefore, those with greater access to marketplaces, service providers and basic resources like ice, sunshade, weighing scales are at a significant advantage to those without.

In addition, traders without reliable supply or credit contracts may experience unfair price fluctuations, poor quality of stock selection or restrictive lending terms [14]. Again, evidence has shown these market relations differ significantly between women and men, as more women retailers report paying higher prices per kilogram to creditors and receiving shorter repayment periods [15]. Research also points to gender-based constraints relating to the personal safety of women retailers [16–19], their limited freedom of mobility [16,18], as well as issues relating to household division of labour, related time burdens [20,21] and unequal distribution of family income and assets [22].

## Conceptual framework

Two major arguments have been presented for integrating gender in value chain research. The first argument builds on evidence showcasing the link between greater gender equity and poverty reduction due to outcomes in improved productivity and innovation, household income, nutrition, child health and education [23,24]. Another argument is based on the link between greater gender equity and enterprise efficiency as a result of fairer allocation of household labour and resources [25,26].

With renewed interest in integrating gender research into global development agenda, an increasing emphasis is on analysing the social institutions that perpetuate gender inequality in economic settings [27–31]. This new agenda moves beyond past frameworks that considered gender as an individual characteristic of women and the potential 'gender gaps' between them [32,33]. Such new frameworks, which adopt a more 'relational perspective', also question the conventional idea of households as a unified economic unit. They emphasize on examining households' internal institutions and power dynamics that govern decision-making and allocation of resources [34].

Limited literature is available on gender in urban fish markets of the Middle East and North Africa region [11,28–31,35–37]. In addition, gender research in aquaculture value chains has tended to focus on upstream nodes of production, with limited studies available of farmed fish retail sectors [11]. This paper addresses this research by also considering novel gender perspectives in the new development agenda laid out above. By adopting a 'social relations approach' [38–45], this study examines the economic significance of gender relations on the trading floor and within the household, which govern how financial resources, labour and market exchange relations, and freedom of mobility are negotiated between women and men. In particular, the study examines the intersection between time use, household and market exchange relations with livelihood outcomes, by cross-referencing our data with age, education, retailing

experience, marital status, household size, location and credit arrangements. As an indicator of livelihood outcome, we focused on the scale of retail enterprises in terms of daily volumes of fish sales. The study explored such gender-based market constraints in three governorates in Lower Egypt. The study's key questions include:

1. What are the household characteristics of informal retailers and do these differ by gender of the retailer?

2. What are the enterprise characteristics of informal retailers and do these differ by gender of the retailer?

3. How do gender-based constraints affect the scale of retail enterprises?

## Methods: Data collection and analysis

This paper focuses on findings from a study in three Lower Egypt governorates of Sharkhia, Kafr el-Sheikh and Beheira. These governorates were selected because of reported concentration of informal retailing activity conducted women traders. The study was carried out between April of 2013 and October of 2014

The total sample size for the three governorates was 345 households, comprising 162 women and 183 men retailers. The sample size was driven by estimates of the population of women and men fish retailers in each governorate, provided by local community development centres. Women fish retailers were then randomly sampled through lists developed by the community development authorities involved in implementing the IEIDEAS project with women retailers: men were identified through lists developed in known fish market places. In Sharkhia, the community development authority's list of women fish retailers only had 30 fish retailers. All these 30 women retailers were included in the study. In Kafr el-Sheikh, there were fewer men fish retailers. Therefore, the total number of interviewed men was only 49 instead of the desired 70. The questionnaire was translated from English into Arabic and field tested in two governorates before being finalized for field implementation.

A structured questionnaire was used to collect data on retailers' household and intra-household characteristics and dynamics in retailing operations and market relations, as well as prices, margins and profits. Tables 1–3 provide the description of variables used in our descriptive and empirical analyses. Field survey activities of the current study were authorized by the Egyptian Ministry of Social Solidarity.

**Table 1. Retailers' household characteristics and dynamics.**

| Variable name | Description | Variable type |
|---|---|---|
| **1.** Sex | Male (0), Female (1) | Dummy |
| **2.** Marital status | Married (0), single (1), widowed/divorced (2) | Categorical |
| **3.** Retailer & spouse age | Years | Continuous |
| **5.** Retailer & spouse education | Years of education | Continuous |
| **6.** Retailer experience | Retailer has more than three years' experience | Dummy |
| **7.** Household size | Number of household members (family/non-family) | Continuous |
| **8.** Household asset index | HH consumer durable asset index including land or vehicle | Continuous |
| **9.** Time in retail | Hours spent in retailing activities; continuous variable | Continuous |
| **10.** Time in household work | Hours spent in household work; continuous variable | Continuous |
| **11.** Help at home | Retailers who 'have family helping with domestic work' | Dummy |
| **12.** Agency in decisions | Own, shared, spouse's decision regarding income control | Categorical |
| **13.** Location | Sharkhia, Kafr el Sheikh, Beheira governorates | Categorical |

**Table 2. Retailers' operations and market characteristics.**

| Variable name | Description | Variable type |
|---|---|---|
| **1.** Market access | Retailers selling in 'marketplace', 'on street' or 'doorstep' | Categorical |
| **2.** Retailer species sales | Retailers selling tilapia, catfish, mullet, or other fish* | Categorical |
| **3.** No. of species sold | Species counted: tilapia, catfish, mullet, carp, mackerel, sardines | Continuous |
| **4.** Cash purchases | Retailers who purchased fish stocks with cash | Dummy |
| **5.** Relations with the creditor | Retailers who reported higher prices when purchasing with credit | Dummy |
| **6.** Loan period | Number of days retailers given to repay loan to creditor/supplier | Continuous |
| **7.** Regular supplier agreement | Retailers who purchase from same supplier daily or weekly | Dummy |
| **8.** Regular buyer agreement | Retailers who 'normally sell to the same buyer' | Dummy |
| **9.** Relations with the supplier | Supplier transports stocks directly to 'point of sale' | Dummy |
| **10.** Fixed & variable costs | Costs floor fees, ice, wages, storage, fines/confiscations (E£) | Continuous |

* Other fish refers carp, mackerel, and sardine species

In Table 1, we present household characteristics and intra-household dynamics of retailers' families. Most of the variables are self-explanatory. We explain in depth the additional variables constructed for this analysis. To examine the wealth status of women and men retailers' households, an 'asset index' was generated. This index was based on ownership of 14 assets (0/1), including colour television, satellite dish, landline telephone, mobile phone, electric fan, air conditioner, refrigerator, freezer, automatic washing machine, manual washing machine, bicycle or motorcycle, animal drawn cart, larger vehicle (car/van/truck/tuk-tuk) and agricultural land. The index was then calculated based on an unweighted average of scores across all assets.

Next, we assess the time burdens of or constraints faced by the retailers, using the survey information on family members' labour roles and hours spent in different paid and unpaid activities. For time-use, we asked the respondents to provide estimates of hours spent in retailing activities, and unpaid household chores and care provision. To explore time burdens further, we asked the retailers whether they experienced 'time conflict' between paid and unpaid activities. Time constraints and work burden depend on the number of household members who demand as well as can contribute labour in household and care provision activities. We constructed 'help at home' variable, which indicates whether or not a retailer has adult members whose primary occupation is household and care work. The intuition behind this variable is to explore how the participation of other household members in household and care responsibilities relates to fish retailers' market work performance.

**Table 3. Retailers' sales and profit margins.**

| Variable name | Description | Variable type |
|---|---|---|
| **1,** Buying price | Purchase prices for tilapia, catfish, mullet, other fish (E£/kg) | Continuous |
| **2.** Morning sales price | Morning sales prices for tilapia, catfish, mullet, other fish (E£/kg) | Continuous |
| **3.** Evening sales price | Evening sales prices for tilapia, catfish, mullet, other fish (E£/kg) | Continuous |
| **4.** % sold at reduced price | Share of stock sold at reduced price (at evening price) | Ratio/Scale |
| **5.** Total volume bought & sold | Total daily volume of all species bought & sold (kg/day) | Continuous |
| **6.** Total value sold | Total daily value of all species sold | Continuous |

We also assess retailers' agency over household finance, for which the survey asked the respondents about intra-household decision-making dynamic in the use of their earnings. Specifically, retailers were asked if they 'only' made the decision, or their 'spouse only' made the decision or 'they and their spouse jointly' made the decision.

In Table 2, we present the variables related to retailers' operations and market characteristics. The variables such as market access, type of fish sold and number of species sold, give the information on the informal retailers' trading operations while variables 5–9 show the mechanisms of retailers' relationships with buyers and sellers. Variables 4 and 10 indicate financial health of informal fish retailers, e.g., whether they are able to purchase fish using cash or credit, and how much they are able to invest in the capital (fixed and variable). Table 3, on the other hand, provides all the information of trading operations like buying and selling volumes and prices.

For data analysis, we employ mean tests, analysis of variance using one-way ANOVA and t-tests across the sample disaggregated by sex and location. For sex-disaggregated analysis between women and men, t-tests were used and for between more than two subsamples, such as location, ANOVA tests were conducted. Linear regression models with robust estimates were employed to determine the direction and intensity of association of predictor variables on scale of enterprise (total value of daily sales). In our model, we examined how household gender dynamics, proxied by time-use in fish retail and unpaid care work, and family help with household work were associated with men and women's scale of fish retailing differently, if at all. The model also evaluated the associations of market-based factors on retailing performance, by examining the significance of trade relations, access to supply markets and enterprise characteristics including variety of species sales, fixed and variable costs.

The regression model, based on the ordinary least squares (OLS) method, was specified as follows:

$$Y_i = \alpha_0 + \alpha_1 S_i + \beta tGBC + \gamma RC + \nu TC + \lambda CO + \varphi_1 R_1 + \varphi_2 R_2 + \mu_i$$

where $Y_i$ is the performance indicator, (scale of operation) of retailer $_i$, $S_i$ is the sex of the retailer; $GBC$ is the vector of gender-based constraints like time spent in fish retail, household care work, availability of help with household chores, interaction of household help with the sex variable; $RC$ is the vector of retailers' characteristics like age, education and years of experience in fish retail; $TC$ is the vector of retail related variables like fixed and variable costs, number of fish species traded, and certainty of supply relations (buying from the same supplier and location of supply transactions); $CO$ is the vector of control variables including regional dummies ($R_1$ and $R_2$); and $\mu_i$ is the error term. The model specifies interaction terms between the proxies for gendered constraints and sex of the retailer primarily because we hypothesize that the effects of these constraints are very different for men and women due to patriarchal norms. We analyze this model for the dependent variable: scale of operation. We measured the scale of enterprise as a natural logarithm of total value of all fish sales. The OLS estimates of the model are presented in Table 7.

The data were tested for violation of normality and equality of variance assumption applying the Breusch-Pagan and Cool Weisberg test, which yielded a $X^2$ of 0.550 suggesting homoscedasticity that is consistent with assumption of linear regression. Post-residual analysis was conducted using the Shapiro-Wilk's test, which yielded insignificant results (p = 0.112) suggesting the assumption of normality was met.

## Results

### Retailers' household characteristics and social relations

Table 4 shows basic descriptive statistics of retailers surveyed. Mean household size was 4.18 ± 1.61. On an average, women retailers were older (37.0 ± 8.2) than men retailers (33.6 ± 10.0) [$F_{(1, 343)}$ = 3.055, p = 0.002]. The number of years of retailer's education also differed significantly by sex [$F_{(1, 3430)}$ = 6.208, p = 0.000], with men retailers reporting a higher educational attainment (8.7 ± 7.0) than that of women retailers (4.3 ± 6.1). We also found that women in the male retailers' households received higher education than women in the female retailers' households, although this difference was not significant for all governorates. Conversely, men in male retailers' households were more educated than men in the female retailers' households and this difference was statistically significant across all governorates.

Across total sample, we observed a significant gender difference in household asset values as shown by the asset index [$F_{(1, 343)}$ = 3.965, p = 0.000]. As an indicator of wealth, we compare household asset index calculated based on ownership of land and consumer durables. Overall, we found that men retailers have a significantly higher asset index compared to that of women. In governate sub-samples, it was statistically significant only in Beheira [$F_{(1, 132)}$ = 3.638, p = 0.000].

We explored four different indicators of intra-household labour allocation–time-use in paid and unpaid activities, whether or not retailer faced time conflict and availability of household members' help in unpaid care work. Overall, we did not find significant difference in time spent in fish retailing across men and women retailers, except in the governate of Sharkhia. Conversely, across the sample, women retailers spent significantly more hours in household work and childcare (5.3 ± 2.3) than men did (3.5 ± 1.3) [$F_{(1, 343)}$ = -9085, p = 0.000]. This difference was statistically significant in all the governorates.

Overall, there were significant gender differences in 'time conflict' variable [$F_{(1, 343)}$ = 7.524, p = 0.000]. Compared to men, a significantly higher proportion of women retailers reported that they face time conflicts between retail work and household responsibilities. This difference was also statistically significant in all governorates. Regarding intra-household

**Table 4. Retailer's household characteristics and dynamics.**

|  | Sharkhia | | | | Kafr el-Sheikh | | | | Beheira | | | | Total | | | |
|---|---|---|---|---|---|---|---|---|---|---|---|---|---|---|---|---|
|  | Female | Male | All | sig# | Female | Male | All | sig# | Female | Male | All | sig# | Female | Male | All | sig# |
| N | 30 | 68 | 98 |  | 64 | 49 | 113 |  | 68 | 66 | 134 |  | 162 | 183 | 345 |  |
| Retailer married (%) | 0.83 | 0.78 | 0.80 |  | 0.73 | 0.79 | 0.74 |  | 0.69 | 0.70 | 0.69 |  | 0.73 | 0.75 | 0.74 |  |
| Retailer age (yrs) | 33.80 | 32.57 | 32.95 |  | 36.19 | 37.78 | 36.88 |  | **39.07** | 31.55 | 35.37 | **** | **36.96** | 33.60 | 35.17 | *** |
| Retail experience (>3yrs, %) | 0.57 | 0.63 | 0.61 |  | 0.83 | 0.76 | 0.80 |  | 0.94 | 0.91 | 0.93 |  | 0.83 | 0.77 | 0.79 |  |
| Retailer education (yrs) | 5.10 | **9.82** | 8.38 | *** | 5.17 | **8.57** | 6.65 | *** | 3.15 | **7.70** | 5.39 | **** | 4.31 | **8.72** | 6.65 | **** |
| Spouse age (yrs) | 36.52 | 30.57 | 32.74 | *** | **45.31** | 33.56 | 39.99 | **** | **43.72** | 28.87 | 36.28 | **** | **42.84** | 30.85 | 36.40 | **** |
| Spouse education (yrs) | 3.11 | 2.40 | 2.88 |  | **3.20** | 2.54 | 2.85 |  | **2.85** | 2.13 | 2.48 | ** | **3.05** | 2.35 | 2.73 | *** |
| Household size | 4.50 | 4.18 | 4.28 |  | 4.53 | 4.10 | 4.35 |  | 3.74 | 4.23 | 3.98 | * | 4.19 | 4.17 | 4.18 |  |
| Household asset index score | 5.83 | 6.01 | 5.96 |  | 5.09 | 5.22 | 5.15 |  | 5.46 | **6.12** | 5.78 | **** | 5.38 | **5.84** | 5.63 | **** |
| Time in retail (hrs/day) | 7.8 | **8.5** | 8.3 | ** | 10.0 | 9.9 | 10.0 |  | 10.0 | 10.0 | 10.0 |  | 9.6 | 9.4 | 9.5 |  |
| Time in hh work (hrs/day) | **6.0** | 3.6 | 4.9 | **** | **3.9** | 3.3 | 3.7 | * | **6.0** | 3.6 | 4.9 | **** | **5.3** | 3.5 | 4.4 | **** |
| Time conflicts with retail | **0.63** | 0.24 | 0.36 | **** | **0.70** | 0.31 | 0.53 | **** | **0.59** | 0.27 | 0.43 | **** | **0.64** | 0.27 | 0.44 | **** |
| Help at home | 0.17 | **0.91** | 0.68 | **** | 0.48 | **0.76** | 0.60 | *** | 0.28 | **0.88** | 0.57 | **** | 0.34 | **0.86** | 0.61 | **** |
| Agency in income decisions (%) | 0.20 | **0.98** | 0.74 | **** | 0.30 | **0.98** | 0.60 | **** | 0.56 | **1.00** | 0.77 | **** | 0.39 | **0.99** | 0.71 | **** |

# Significant p-values are reported at levels of <0.10, <0.05, <0.01, <0.001 as *, **, ***, ****.

labour roles, we constructed a 'help at home' variable to assess if retailers had family help in household tasks and care provision. Across the sample, a significantly higher share of men reported having family help in the home (85.8%) than women did (34.0%). This is obvious due to cultural constructs of patriarchy. Often men have their wives taking care of household chores and care work while for women retailers, unless they have older daughters or other women in the household, they seldom receive their spouse's help in these responsibilities.

With regards to intra-household decision-making related to control and use of income, a significantly higher percentage of men reported 'self' as opposed to their spouse as the prevailing decision-maker (98.8%). Only 39.2% of women reported themselves as the prevailing decision-maker. This difference was also statistically significant in every governorate. This result indicates that even when women have a source of livelihood, they seldom make decisions about how to use that income, in other words, women have to consult or report their husbands on the use of income.

## Retailers' operations

In Table 5 below, we present the descriptive results of indicators related to retailing aspects of informal fish traders in Egypt. Across the aggregate sample, a higher share of men retailers reported access to formal 'marketplaces' (91.3%) than women retailers did (85.2%). In contrast, significantly more women reported selling fish on their 'doorstep' (5.6%) than men did (0.6%) [$F(1, 343) = -2.791, p = 0.006$]. This difference was highly significant in Sharkhia, where 30% of women sold fish on their 'doorstep' while no men did so. While both men and women in this sample are informal retailers, this result indicates the higher intensity of informality faced by women retailers. The lack of access to a formal marketplace for women and selling in front of the house imply that these women retailers do not have the opportunity to expand their clientele. It also indicates the constraints faced by women in working away from home as they often mange their household responsibilities along with market work.

**Table 5. Retailers' operations and market characteristics.**

| | Sharkhia | | | | Kafr el-Sheikh | | | | Beheira | | | | Total | | | |
|---|---|---|---|---|---|---|---|---|---|---|---|---|---|---|---|---|
| | Female | Male | All | sig# | Female | Male | All | sig# | Female | Male | All | sig# | Female | Male | All | sig# |
| *N* | *30* | *68* | *98* | | *113* | *49* | *113* | | *68* | *66* | *134* | | *162* | *183* | *345* | |
| Retailer trades in marketplace (%) | 0.70 | 0.93 | 0.86 | *** | 0.78 | 0.78 | 0.78 | | 0.99 | 1.00 | 0.99 | | 0.85 | **0.91** | 0.88 | * |
| Retailer trades on street (%) | 0.00 | 0.07 | 0.05 | | 0.22 | 0.20 | 0.21 | | 0.01 | 0.00 | 0.01 | | 0.09 | 0.08 | 0.09 | |
| Retailer trades on doorstep (%) | **0.30** | 0.00 | 0.09 | * | 0.00 | 0.02 | 0.01 | | 0.00 | 0.00 | 0.00 | | **0.06** | 0.01 | 0.03 | *** |
| Retailer sells tilapia (%) | 0.97 | 0.82 | 0.87 | * | 0.94 | 0.96 | 0.95 | | **0.94** | 0.79 | 0.87 | *** | **0.94** | 0.85 | 0.89 | *** |
| Retailer sells catfish (%) | 0.20 | 0.15 | 0.16 | | 0.27 | 0.31 | 0.28 | | **0.43** | 0.12 | 0.28 | **** | **0.32** | 0.18 | 0.25 | *** |
| Retailer sells mullet (%) | 0.00 | **0.19** | 0.13 | | 0.67 | 0.71 | 0.69 | | 0.12 | 0.15 | 0.13 | | 0.31 | 0.32 | 0.32 | |
| Retailer sells other fish (%) | 0.37 | 0.44 | 0.42 | | 0.16 | 0.14 | 0.15 | | 0.12 | **0.45** | 0.28 | **** | 0.18 | **0.37** | 0.28 | **** |
| No. of species sold | 1.5 | 1.7 | 1.6 | | 2.1 | 2.2 | 2.1. | | 1.6 | 1.8 | 1.7 | | 1.8 | 1.9 | 1.8 | |
| Supplier paid w/ cash (%) | 0.43 | 0.51 | 0.49 | | 0.28 | **0.49** | 0.37 | ** | 0.06 | **0.18** | 0.12 | ** | 0.22 | **0.39** | 0.31 | **** |
| Creditor increases prices (%) | 0.12 | 0.15 | 0.14 | | **0.28** | 0.05 | 0.21 | ** | 0.11 | 0.15 | 0.13 | | 0.17 | 0.13 | 0.15 | |
| Loan period (days) | 1.2 | 1.5 | 1.5 | | 2.7 | 3.0 | 2.8 | | 1.4 | 1.8 | 1.5 | | 1.8 | 2.0 | 1.9 | |
| Regular supplier agreement (%) | 0.80 | 0.63 | 0.68 | | 0.33 | 0.39 | 0.35 | | 0.82 | 0.71 | 0.77 | | 0.62 | 0.60 | 0.61 | |
| Regular buyer agreement (%) | 0.50 | 0.31 | 0.37 | * | 0.16 | 0.14 | 0.15 | | **0.35** | 0.06 | 0.21 | **** | **0.30** | 0.17 | 0.23 | *** |
| Supplier travels to retailer (%) | **0.97** | 0.81 | 0.86 | ** | 0.91 | 0.90 | 0.90 | | 0.69 | 0.94 | 0.81 | **** | 0.83 | 0.88 | 0.86 | |
| Fixed & variable costs (E£/month) | 23.1 | **61.7** | 49.9 | *** | 104.8 | 122.1 | 112.3 | | 39.1 | **62.8** | 50.8 | *** | 62.1 | **78.3** | 70.7 | * |

# Significant p-values are reported at levels of <0.10, <0.05, <0.01, <0.001 as *, **, ***, ****.

Across the sample, tilapia was the most commonly sold species by retailers (89.3%), followed by mullet (31.6%) and catfish (24.6%). 'Other fish' category included carp (*Labeo niloticus*, Forsskål, 1775), mackerel (*Scomberomorus* spp., Lacepède, 1801), and sardines species (*Sardinella* spp., Valenciennes, 1847).

It is important to note the price differentials between these species. Due to Egypt's rapid aquaculture development, the cheapest available animal-source protein in retail markets is now farmed Nile tilapia (*Oreochromis niloticus*, Linneaus, 1758), followed by catfish (*Clarius gariepinus*, Burchell, 1822; *Heterobranchus bidorsalis*, Saint-Hilaire, 1809; *Schilbe mystus*, Linnaeus, 1758; *Bagrus* spp., Bosc, 1816). Other species, including mullet (*Mugil* spp. Linnaeus, 1758), are sold at higher prices, often in higher value markets. With this in mind, we note that a higher percentage of women retailers were trading in lower value products. Overall, we found significant differences between men and women retailers' sales of tilapia sales, catfish sales and other fish sales. On the one hand, higher proportions of women sold tilapia (94.4%) and catfish (32.1%) than men did (84.7%, 18.0%, respectively). On the other, a significantly higher percentage of men sold other fish (36.6%) than women did (17.9%). While overall no statistically significant difference was observed in mullet sales by sex, we did find, however, in Sharkhia that no women reported selling this species.

Considering exchange and credit relations of retailers here, we find that only 30.7% of respondents used cash for purchasing fish stocks, while significantly fewer women retailers (21.6%) than men retailers used cash (38.8%). Instead, a majority of retailers reported borrowing credit from either suppliers (67.2%), fishers (2.9%) and family (0.5%). For retailers using credit, 15.3% reported having to pay higher prices for stock due to loan agreements. This was most significant for women in Kafr el-Sheikh where 28.3% reported price hikes when buying fish with credit compare to 4.5% of men retailers. Considering the differences in loan repayment periods, we found that significant differences across governorates, with greatest difference found between Beheira (2.81 days ± 2.06) and Sharkhia (1.48 days ± 1.38).

Asking retailers about their upstream and downstream trade relations, we discussed their agreements and transactions with buyers or suppliers who mainly operated as men distributors trading and transporting fish between farming locations or wholesale auction houses and retail markets or retailer households. We observed statistically significant differences between men and women retailers in 'regular buyer agreement'. Across the sample, higher percentage of women reported selling fish to the same buyers (30.2%) than men did (17.5%). This difference was also statistically significant between women (35.3%) and men (6.1%) in Beheira governorate. Considering retailers' access and travel to supply markets, we asked retailers if suppliers travelled directly to their point of sale. We observed statistically significant differences between men and women retailers in Sharkhia and Beheira. While in Sharkhia more women reported point of sale supplier (96.7%) than men did (80.9%), in Beheira, fewer women reported point of sale supplier.

## Retailers' sales and profit margins

In Table 6, data on previous day of retailers' fish purchases, sales, and monthly variable and fixed costs are presented, in addition to gross margins for each species and daily gross profits for total stock. In Sharkhia, Beheira, and for the overall sample, gross margin of tilapia per kilogram (i.e. price sold minus price bought) was significantly higher for women than for men. Across locations, highest mean margin for a kilogram of tilapia sold was in Beheira (E£1.65 ± 2.88) and lowest in Sharkhia (E£0.63 ± 1.85). For catfish sales in Sharkhia, women reported significantly higher daily margins (E£2.59 ± 0.18) than men retailers (E£1.44 ± 0.16) [$F_{(1, 14)} = 5.344$, $p = 0.000$].

**Table 6. Retailers' sales and profit margins.**

| | Sharkhia | | | | Kafr el-Sheikh | | | | Beheira | | | | Total | | | |
|---|---|---|---|---|---|---|---|---|---|---|---|---|---|---|---|---|
| | Female | Male | All | sig# | Female | Male | All | sig# | Female | Male | All | sig# | Female | Male | All | sig# |
| *N* | 30 | 68 | 98 | | 64 | 49 | 113 | | 68 | 66 | 134 | | 162 | 183 | 345 | |
| *Tilapia n* | 29 | 56 | 85 | | 60 | 47 | 107 | | 64 | 52 | 116 | | 153 | 155 | 308 | |
| Buying price (E£/kg) | 9.6 | **10.3** | 10.1 | *** | 9.2 | 9.1 | 9.2 | | 9.9 | **10.5** | 10.2 | ** | 9.6 | **10** | 9.8 | ** |
| Morning sales price (E£/kg) | 11.6 | 11.9 | 11.8 | | 12.3 | 12.1 | 12.2 | | 12.1 | 12.4 | 12.2 | | 12.1 | 12.1 | 12.1 | |
| Evening sales price (E£/kg) | 10.5 | **11.3** | 11.0 | * | 9.9 | 9.8 | 9.9 | | 11.3 | 11.2 | 11.3 | | 10.3 | 10.5 | 10.4 | |
| Share sold at reduced price (%) | 0.10 | 0.09 | 0.10 | | 0.22 | 0.19 | 0.21 | | 0.06 | 0.05 | 0.05 | | 0.13 | 0.11 | 0.12 | * |
| Gross margin (E£/kg) | 1.3 | 0.3 | 0.6 | ** | 0.8 | 1.0 | 0.9 | | **2.1** | 1.1 | 1.6 | * | 1.4 | 0.8 | 1.1 | ** |
| *Catfish n* | 6 | 10 | 16 | | 17 | 15 | 32 | | 29 | 8 | 37 | | 52 | 33 | 85 | |
| Buying price (E£/kg) | 7.8 | **9.1** | 8.6 | ** | 10.5 | 9.6 | 10.1 | | 8.4 | **9.5** | 8.6 | * | 9 | 9.4 | 9.2 | |
| Morning sales price (E£/kg) | 10 | 10.6 | 10.4 | | 13.1 | 12.4 | 12.8 | | 10.5 | 11.1 | 10.7 | | 11.3 | 11.6 | 11.4 | |
| Evening sales price (E£/kg) | 8 | 9.5 | 9.2 | | 11.4 | 9.9 | 10.7 | | 10.6 | 11.5 | 10.8 | | 11 | 10 | 10.5 | |
| Share sold at reduced price (%) | 0.01 | 0.08 | 0.05 | | 0.22 | 0.19 | 0.21 | | 0.05 | 0.07 | 0.05 | | 0.1 | 0.13 | 0.11 | |
| Gross margin (E£/kg) | **2.6** | 1.4 | 1.9 | *** | 2.5 | 2.4 | 2.4 | | 2.1 | 1.8 | 2 | | 2.3 | 1.9 | 2.1 | |
| *Mullet n* | 0 | 13 | 13 | | 43 | 35 | 78 | | 8 | 10 | 18 | | 51 | 58 | 109 | |
| Buying price (E£/kg) | 0 | **15.4** | | | 19 | 18.7 | 18.9 | | **18.5** | 16.7 | 17.5 | * | **18.9** | 17.6 | 18.2 | ** |
| Morning sales price (E£/kg) | 0 | **18.3** | | | 22.8 | 22.1 | 22.5 | | 21.1 | 19.5 | 20.2 | | **22.6** | 20.8 | 21.6 | ** |
| Evening sales price (E£/kg) | 0 | **21.5** | | | 20.1 | 19.4 | 19.8 | | 19.2 | 16 | 18.4 | | 20 | 19.5 | 19.7 | |
| Share sold at reduced price (%) | 0 | 0.03 | | | 0.26 | 0.2 | 0.25 | | 0.02 | 0.07 | 0.05 | | **0.22** | 0.16 | 0.19 | * |
| Gross margin (E£/kg) | 0 | **3.6** | | | 2.1 | 2.7 | 2.4 | | 3.3 | 2.4 | 2.8 | | 2.3 | 2.9 | 2.6 | |
| *All fish* | | | | | | | | | | | | | | | | |
| Total volume bought (kg/day) | 45.7 | **84.7** | 72.7 | *** | 87.8 | **102.1** | 94.0 | | 89.7 | 102.7 | 96.1 | | 80.8 | **95.8** | 88.8 | ** |
| Total volume sold (kg/day) | 46.5 | **81.5** | 70.8 | *** | 88.9 | **103.0** | 95.0 | | 90.9 | 98.5 | 94.6 | | 81.9 | **93.4** | 88.0 | * |
| Total value sold (E£/day) | 481.4 | **935.9** | 769.8 | *** | 1070.3 | **1294.6** | 1167.6 | | 1026.1 | 1172.9 | 1098.4 | | 942.7 | **1117.4** | 1035.4 | ** |
| Gross profit daily (E£/day) | 91.1 | 135.6 | 122.0 | | 192.4 | **234.9** | 210.8 | * | 216.9 | 191.6 | 204.4 | | 183.9 | 182.4 | 183.1 | |

# Significant p-values are reported at levels of <0.10, <0.05, <0.01, <0.001 as *, **, ***, ****.

In all governorates, women retailers reported lower daily volume and value of all fish transactions than their male counterparts. Specifically, results showed sex had significant effect on volume of daily purchases [$F_{(1, 343)}$ = 2.634, p = 0.009], volume of daily sales [$F_{(1, 343)}$ = 2.020, p = 0.044], and value of stock sold daily [$F_{(1, 343)}$ = 2.463, p = 0.014]. Greatest differences between sales of women and men was found in Sharkhia, where, for instance, the value of stocks sold by men retailers (E£935.9 ± 74.8) was almost double the value of stock sold by women (E£481.4 ± 48.5). This reflects also on the diversity of species and different prices of species sold by retailers in this location, as no women reported selling higher value mullet.

## Empirical analysis of social and market-based factors to retailing performance

The current study investigated whether or not there were differences between male and female fish retailers' enterprise performance after controlling for individual, household, regional, business and entrepreneurial characteristics. We also examined the association between gender-based constraints and retail performance, namely variables informing gender dynamics within retailers' households and those considered significant to market operations and exchange relations.

**Table 7. Determinants of scale of enterprise (daily volume of sales).**

| Number of observations = 345 | Enterprise scale# |
|---|---|
| **Retailers' characteristics** | |
| Sex (female = 1) | -0.370 (0.104)**** |
| Age (yrs) | -0.002 (0.003) |
| Retailing experience (yrs) | 0.104 (0.071) |
| Retailer education (yrs) | -0.022 (0.014) |
| **Household characteristics** | |
| Time in retail (hrs) | 0.107 (0.024)**** |
| Time in hh work (hrs) | -0.059 (0.023)*** |
| Help at home (1) | -0.325 (0.100)**** |
| Help at home*Sex | 0.260 (0.127)** |
| **Trading characteristics** | |
| Fixed and variable costs (E£/month) | 0.442 (0.046)**** |
| Supplier travels to retailer (1) | -0.190 (0.072)*** |
| Supplier regular agreement (1) | 0.146 (0.056)*** |
| No. of species sold | 0.269 (0.036)**** |
| **Regional dummies** | |
| Sharkhia (1) | -0.037 (0.074) |
| Kafr el Sheikh (1) | -0.107 (0.072) |
| Constant | 4.707 (0.277)**** |
| **Model evaluations** | |
| $R^2$ | 0.546 |
| Adjusted $R^2$ | 0.527 |
| Breusch-Pagan/Cool Weisberg | 0.550 |
| Shapiro-Wilk | 0.112 |

\# Enterprise scale dependent variable defined as total value of daily sales (E£/day).

Significant p-values are reported at levels of $<0.10$, $<0.05$, $<0.01$, $<0.001$ as *, **, ***, ****.

As described earlier in the framework section, the study employed linear regression method to analyze these associations. Results reported that the regression equation was found to be significant [$F_{(14, 330)} = 28.35$, $p = 0.000$], with an $R^2$ value of 0.546 explaining 53% of the variance in the dependent variable (Adjusted $R^2 = 0.527$). The results, presented in Table 7, revealed positive and highly significant relationships between enterprise performance and the sex dummy (male = 1), higher socioeconomic status of respondent, and time spent in retailing We observed a highly significant association between sex and retailers' scale of operation, with the model predicting that women retailers compared to men retailers sustain smaller scale of enterprise ($p = 0.001$).

To understand the significance of household roles and related time burdens, we examined the effects of time spent in 'retailing', time spent in 'household work', the availability of family 'help at home.' We also created an interaction term between household help at home and the sex dummy in order to separate the effect of this variable by sex. On the one hand, we observed significantly positive associations of increased 'time spent in retailing' with scale of enterprise, on the other, increased 'time spent in unpaid care work' expressed a significantly negative correlation with scale of enterprise. Interestingly, the availability of household help from family members for both men and women exhibited negative association with the dependent variable ($p = 0.001$). However, considering the interaction term between household help and sex, we found that household help was significantly and positively associated with women's scale of

enterprise. This implies that among women retailers who have other household members sharing the burden of chores and care work, the scale enterprise is significantly higher. The study's descriptive analysis can shed further light on these results (Table 4). Across the aggregate sample, 62% of retailers had one or more household members who are able to shoulder the responsibilities of household chores and care provision. However, between men and women retailers, the difference was stark: 85% vs. 34%, respectively. The men retailers who do not have household help from other members are primarily single males with smaller household size. However, in case of women without household help, the majority still bears the burden of managing a family and providing care to her spouse and children. To be able to comment more accurately on the interactions of these phenomenon with different aspects of retailing performance require further exploration, which is beyond the scope of the current study. However, in the final section, we discuss the literature exploring these dynamics in other economic spheres.

We did not find any association between enterprise scale and other individual level characteristics, namely, age, experience in retailing and education. Not surprisingly, both independent variables 'number of species sold' and 'operational costs', expressed significantly positive associations with the outcome variable ($p < 0.001$). These variables indicate the retailer's performance and investment capability. Those who are able to invest more in fixed capital as well as the variable expenses of fish retailing tend to manage a bigger scale of operations. Also, a wider variety of fish species attracts more customers and enables generation of higher revenues.

The mechanisms of the supply markets were also significant to scale of enterprise, but in surprising ways. One the one hand, the retailers who had a verbal or written contract with their supplier, were generating more revenues or operating larger scale of enterprise. This result indicates that an agreement with the supplier may ensure stability in the supply of fish for sale and goodwill, thereby, may facilitate higher sales. For the retailers whose suppliers travelled to their location of retail point of sale, there was a negative association of this aspect with the scale of enterprise. It may be possible that in these cases, the supplier is selling leftover poor quality fish to the retailers and may not deliver timely, therefore, impeding revenue generation ability of the retailers.

## Conclusion

In Egypt, which is a strong patriarchal society, we expect gender-based constraints and the social norms that underpin them to affect men and women differently. The two sets of literature from feminist economics and informal markets suggest two different categories of factors that affect the performance of informal retail businesses–retailers' household aspects and intra-household dynamics, and enterprise characteristics [44]. Evidence from the literature also shows that after controlling for these factors, retailer's sex is highly significant in determining business performance and market transactions [42,43,45–49]. Our findings confirm this hypothesis and present empirical evidence for Middle Eastern region.

The results of this study suggest that sex of retailer proves significant to scale of enterprise, as do household and trading characteristics (Table 7). In particular, the regression and descriptive results indicate that as time spent in retailing increased, retailers' scale of enterprise increased. Conversely, for those with greater burden of household care and unpaid activities (as indicated by time-use in these activities), the scale of enterprise was lower. Given that gendered household labour relations mean that domestic time burdens fall disproportionately on women, these gendered relations also disproportionately negatively impact the business performance of women compared to men. Our results confirm similar findings to those reported

by Arora's study of peasant households in Mozambique [38,39,50–59] Arora and Rada, 2016). That is, labour allocation of unpaid activities such as child-care and domestic chores produce greater time burdens on women; as well as affect individual economic activities, which likely influences overall household productivity in terms of livelihood security and gross profitability.

At the market scale, our findings also indicate that the availability of supply agreements were significant to trade performance (Table 7). These findings echo similar discussions by Sabarwal and Terrell (2008), in which they highlighted capital constraints and limited access to financial markets as key barriers to women's enterprise growth. It also reflects on findings above which point to the market-based dynamics affected gendered bargaining power and the gender-differences identified in scale of daily sales by retailers. This reflects also on the differences observed between reported operational costs of women and men. Across the sample, men reported higher operational costs, while results on type and number of species sold also indicated that men were predominantly involved in the sale of higher value products (Table 5). Considering findings in the literature and the results of this study on gendered bargaining power in the economic sphere, including differential access to supply and financial markets, we find important spaces of gender inequality worthy of investigation that evidently affect gender differences in enterprise outcomes.

Considering the significance of supply relations, results also indicated that retailers relying point of sale wholesalers suffered poorer scale of enterprises. This raises some important questions for future research, which calls for more detailed analysis of supply markets and the transactions between retailers and wholesalers, both within the local markets and within regional wholesale depots. Evidently, there are important differences between such spaces regarding women's bargaining power in these male dominated marketplaces. In Egypt, several studies have highlighted varied constraints facing women's retailing enterprises, namely the lack of supply contracts, restrictive lending terms and unfair price and trade negotiations, all of which stem in part from vulnerable credit relations or trade agreements [11,15]. These combined factors add pressure to the risk of daily spoilages and the need to clear daily stock before market closes, therefore, placing further pressure on retailers' bargaining power, all of which were made evident by changing sales prices and differences in profit margins throughout the day (Table 6).

Such gender-based market constraints should be considered in future research through more detailed investigation of time-use variables and intra-household decision-making between dual respondents between dual-adult and female-headed households. Literature on the gendered dynamics of intra-household decision-making have examined their significance to household economics, agricultural productivity as well as off-farm paid market activities [21,39]. While our survey collected data on household structure among both single and dual-adult households, the percentage of single-adult households and the sampling only of lead respondents did not allow for intra-household analysis at the depth required. Therefore, we recommend future research to adopt more purposive sampling of dual-adult respondents that allows for robust investigation of household decision-making processes.

Finally, the dynamics identified as significant in this study imply the need for more detailed analysis of how social institutions and gender norms in both the marketplace and within the household affect livelihood options and outcomes of women retailers. Ignoring such gender dynamics may impinge on enterprise performance, scale and profitability of retailing operations among women and thus, contribute to perpetuating the cycle of poverty amongst the urban poor. Moreover, these dynamics may limit the ability of women retailers to affordably supply the fish needed to meet the growing demands in these critical markets for poor, food insecure, consumers. To address such market constraints and improve the agency and

bargaining power of female economic actors, value chain interventions require approaches that engage with both men and women retailers from within household and across the community. We offer two main recommendations: 1) interventions of gender-responsive value chain interventions retail should consider the findings of such current studies to propose policy strategies that specifically address enabling factors such as sharing of household labour in paid and unpaid labour, and the strengthening of bargaining power of women retailers in key spaces of transactions and access to physical supply and financial markets. 2) for effective implementation of gender-responsive market intervention, we recommend that initiatives should consult with and deliberate the voices of both women and men retailers and community stakeholders to ensure that women maintain a voice in policy and technology interventions.

## Supporting information

**S1 Syntax.**
(DO)

**S2 Syntax.**
(DO)

**S3 Syntax.**
(DO)

**S1 Dataset.**
(DTA)

## Acknowledgments

The data in this study were collected in the context of the project "Improving Employment and Income through Developing the Egyptian Aquaculture Sector project (IEIDEAS). We thank all donors who supported this work. We would like to thank North South Consultants Exchange for their work to collect the data, and CARE Egypt for its support to access IEIDEAS project beneficiaries and for providing thoughtful comments on an earlier draft. Finally, we are grateful for the inputs from Malcolm Dickson and Ihab Anwar to this paper.

## Author Contributions

**Conceptualization:** Froukje Kruijssen, Cynthia McDougall, Paula Kantor.

**Data curation:** Seamus Murphy, Froukje Kruijssen, Cynthia McDougall, Paula Kantor.

**Formal analysis:** Seamus Murphy, Diksha Arora.

**Funding acquisition:** Cynthia McDougall, Paula Kantor.

**Investigation:** Froukje Kruijssen, Paula Kantor.

**Methodology:** Seamus Murphy, Froukje Kruijssen, Cynthia McDougall.

**Project administration:** Cynthia McDougall.

**Resources:** Seamus Murphy, Froukje Kruijssen.

**Software:** Seamus Murphy.

**Supervision:** Seamus Murphy, Froukje Kruijssen, Cynthia McDougall.

**Validation:** Diksha Arora, Cynthia McDougall, Paula Kantor.

**Writing – original draft:** Seamus Murphy, Froukje Kruijssen.

**Writing – review & editing:** Seamus Murphy, Diksha Arora, Cynthia McDougall.

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
