## [Decision Letter · Decision Letter 0]

23 Sep 2019

PONE-D-19-18427

Gender-based market constraints to informal fish retailing; Evidence from analysis of variance and linear regression

PLOS ONE

Dear Dr Murphy,

Thank you for submitting your manuscript to PLOS ONE. After careful consideration, we feel that it has merit but does not fully meet PLOS ONE’s publication criteria as it currently stands. Therefore, we invite you to submit a revised version of the manuscript that addresses the points raised during the review process.

We recommend that it should be revised taking into account the changes requested by the reviewers. Since the requested changes includes Minor Revision, the revised manuscript will undergo the next round of review by the same reviewers.

We would appreciate receiving your revised manuscript by Nov 07 2019 11:59PM. To enhance the reproducibility of your results, we recommend that if applicable you deposit your laboratory protocols in protocols.io, where a protocol can be assigned its own identifier (DOI) such that it can be cited independently in the future. For instructions see: http://journals.plos.org/plosone/s/submission-guidelines#loc-laboratory-protocols

We look forward to receiving your revised manuscript.

Kind regards,

Baogui Xin, Ph.D.

Academic Editor

PLOS ONE

Journal Requirements:

2. Please note that some authors are affiliated with WorldFish, "an international, nonprofit research organization that harnesses the potential of fisheries and aquaculture to reduce hunger and poverty" (https://www.worldfishcenter.org/). As a non-profit research organization whose mission relates to the topic of the study, this should be discussed in the competing interests statement to ensure transparency.

3. Thank you for including your ethics statement on the online submission form:  'Field survey activities of the current study were authorized by the Egyptian Ministry of Social Solidarit'

To help ensure that the wording of your manuscript is suitable for publication, would you please also add this statement at the beginning of the Methods section of your manuscript file.

Additional Editor Comments (if provided):

Reviewers' comments:

Reviewer's Responses to Questions

**Comments to the Author**

1. Is the manuscript technically sound, and do the data support the conclusions?

Reviewer #1: Yes

Reviewer #2: Yes

2. Has the statistical analysis been performed appropriately and rigorously? 

Reviewer #1: Yes

Reviewer #2: Yes

3. Have the authors made all data underlying the findings in their manuscript fully available?

Reviewer #1: No

Reviewer #2: Yes

4. Is the manuscript presented in an intelligible fashion and written in standard English?

Reviewer #1: Yes

Reviewer #2: Yes

5. Review Comments to the Author

Reviewer #1: There is a need to bring out/separate the constraint result. The enterprise and household characteristics seem to be the constraints or the title could be rephrased as household and enterprise constraints of men and women retailers...... The regression is satisfactory, however, I suggest the author takes a second look at the ANOVA/t-test. The measurement of variables is faulty on some variable hence, I doubt some result of on test of difference.

Reviewer #2: In this paper of “ Gender-based market constraints to informal fish retailing. I have several reservations about this paper and which are summarized as follows:

1. The paper has no driving research question. It is not clear to explain the contribution of this paper. In other words, the purpose of this research should be more clearly communicated in the introduction section. In introduction, please review the previous studies. What is the novelty for this article compared with existing studies?

2. I would like to see more discussion of the literature so that I can clearly identify the article relates to competing ideas.

3. authors had better to polish the manuscript.

6. PLOS authors have the option to publish the peer review history of their article (what does this mean?). If published, this will include your full peer review and any attached files.

Reviewer #1: Yes: Olaosebikan Olamide Deborah

Reviewer #2: No

---

## [Author Response · Author response to Decision Letter 0]

31 Jan 2020

Dear Reviewer I,

Thank you very much for reviewing our manuscript (PONE-D-19-18247) entitled “Gender-based market constraints to informal fish retailing; Evidence from analysis of variance and linear regression” for consideration in Plos ONE. Your comments were excellent and have helped us to strengthen the manuscript. All comments were considered and changes were made accordingly, which we hope met your interpretations and recommendations. Please find below our revisions in response to your comments below.

Kind Regards,

Authors

Comments of & responses to reviewer I:

One general comment is that there is need to further explain the ‘constraint result’ and that this explanation needs to detail explicitly what is meant between household- and enterprise-based constraints. In particular, the reviewer suggests that such gender based constraints could separated from such results. 

- The authors appreciated these comments. On p. 3 and 4, l. 56-99, we considered your suggestion by adding information to the introduction and conceptual framework regarding the paper’s focus on and definition of ‘gender-based market constraints’ to accessing and utilizing markets and household resources. We add that, by adopting a ‘social relations approach’, the paper investigates the ‘economic significance’ of gendered power relations both ‘on the trading floor and within the household where financial resources, household labour and market exchange relations are negotiated…’. In response to the reviewer’s further comment regarding a potential separation between gender-based constraints and economic results, we feel that the purpose of the paper is bring attention to the intersections that exist between such enterprise outcomes and gender relations and norms, which are negotiated at both household and market levels. On p. 4, l. 92, in order to emphasize this point, we refer more explicitly to where and how such intersections were investigated, while citing additional literature to help inform the reader of these approaches and debates. 

Another comment is that further methodological discussion is needed of the study’s ANOVA and t-test analyses, which explains the parameters and measurements of variables assessed. 

- The authors appreciated these comments. On p. 7, l. 159-162, we addressed these concerns by adding details regarding how such variables were defined. We also explained more clearly which variables were analyzed using ANOVA methods and which variables were analyzed using t-tests. 

Dear Reviewer II,

Thank you very much for reviewing our manuscript (PONE-D-19-18247) entitled “Gender-based market constraints to informal fish retailing; Evidence from analysis of variance and linear regression” for consideration in Plos ONE. Your comments were excellent and have helped us to strengthen the manuscript. All comments were considered and changes were made accordingly, which we hope met your interpretations and recommendations. Please find below our revisions in response to your comments below.

Kind Regards,

Authors

Comments of & responses to reviewer II:

Two general comments from reviewer II suggest that the paper requires a clearer explanation in the introduction of the paper’s contribution to current literature and of the novelty that the paper offers to such literature. 

- The authors appreciated this comment. On p. 3, l. 79, the authors introduce wider debates regarding the need for integrating gender research in economic development debates. In these discussions, on p. 3-4, l 86-98, the authors highlight how the current paper builds on these perspectives to contribute ‘novel gender perspectives in the new global development agenda’. In particular, the authors outline the study’s aim to provide results using the ‘relational approach’ to value chain research. In addition, the authors highlight how previous gender research in aquaculture retail sectors and value chains within region more generally have been limited, thereby highlighting the of the current study (l. 88). 

A specific comment by the reviewer suggests that the manuscript should be polished. 

- The authors appreciated this comment and have made substantial reviews and edits throughout the paper, restructuring per advice to position methodological discussions into the methods sections and correcting any content where minor mistakes were identified.

---

## [Editor Report · Decision Letter 1]

4 Feb 2020

Gender-based market constraints to informal fish retailing: Evidence from analysis of variance and linear regression

PONE-D-19-18427R1

Dear Dr. Murphy,

We are pleased to inform you that your manuscript has been judged scientifically suitable for publication and will be formally accepted for publication once it complies with all outstanding technical requirements.

With kind regards,

Baogui Xin, Ph.D.

Academic Editor

PLOS ONE
---

## [Editor Report · Acceptance letter]

24 Feb 2020

PONE-D-19-18427R1 

Gender-based market constraints to informal fish retailing: Evidence from analysis of variance and linear regression 

Dear Dr. Murphy:

I am pleased to inform you that your manuscript has been deemed suitable for publication in PLOS ONE. Congratulations! Your manuscript is now with our production department. 

With kind regards,

on behalf of

Prof. Baogui Xin 

Academic Editor

PLOS ONE